# The Cerebellum’s Role in Affective Disorders: The Onset of Its Social Dimension

**DOI:** 10.3390/metabo13111113

**Published:** 2023-10-28

**Authors:** Stefano Stanca, Martina Rossetti, Paolo Bongioanni

**Affiliations:** 1Department of Surgical, Medical, Molecular Pathology and Critical Area, University of Pisa, Via Savi 10, 56126 Pisa, Italy; 2NeuroCare Onlus, 56100 Pisa, Italy; 3Medical Specialties Department, Azienda Ospedaliero-Universitaria Pisana, 56100 Pisa, Italy

**Keywords:** cerebellum, major depressive disorder, bipolar disorder, social impact, neuropsychiatry

## Abstract

Major Depressive Disorder (MDD) and Bipolar Disorder (BD) are the most frequent mental disorders whose indeterminate etiopathogenesis spurs to explore new aetiologic scenarios. In light of the neuropsychiatric symptoms characterizing Cerebellar Cognitive Affective Syndrome (CCAS), the objective of this narrative review is to analyze the involvement of the cerebellum (Cbm) in the onset of these conditions. It aims at detecting the repercussions of the Cbm activities on mood disorders based on its functional subdivision in vestibulocerebellum (vCbm), pontocerebellum (pCbm) and spinocerebellum (sCbm). Despite the Cbm having been, for decades, associated with somato-motor functions, the described intercellular pathways, without forgiving the molecular impairment and the alteration in the volumetric relationships, make the Cbm a new important therapeutic target for MDD and BD. Given that numerous studies have showed its activation during mnestic activities and socio-emotional events, this review highlights in the Cbm, in which the altered external space perception (vCbm) is strictly linked to the cognitive-limbic Cbm (pCbm and sCbm), a crucial role in the MDD and BD pathogenesis. Finally, by the analysis of the cerebellar activity, this study aims at underlying not only the Cbm involvement in affective disorders, but also its role in social relationship building.

## 1. Introduction

Psychiatry, before being the alveus of the diagnosis and therapy of mental disorders, provides, first of all, the interpretative setting for the understanding of human behavior. Indeed, mental disorders represent the exacerbation of the normal cognitive–emotive sphere, whose frame embodies the dimension by which we experience our social dimension. They are *exempla* of a fracture with the world, although, simultaneously, the litmus test of our functioning as social animals, as relational pivot points. Mental disorders are essential in making more evident dynamics normally inherent to human beings. The emotive state, strictly linked to cognition is, in this perspective, the forerunner of theoretical knowledge. On the basis of these premises, in this work, the investigation on the intriguing cerebellum part in emotion is conducted by filtering its potential contribution to social behavior through its role in the pathogenesis of affective disorders. Therefore, in this intertwining of cognition neural correlates and emotion, on the background of normality and pathology between Major Depressive Disorder (MDD; see a list of Abbreviations) and Bipolar Disorder (BD), can the cerebellum (Cbm) acquire a social dimension? What is the role of the Cbm in tracing our harmonic relationship with the otherness and, consequently, its importance in the pathological reaction to environmental stimuli? Accordingly, in this research, affective disorders embody the instrument by which to infer the importance of cerebellum in emotion and, consequently, in social relationships. MDD is characterized by a depressive mood bending lasting longer than 2 weeks. It describes a dramatic scenario in which emotional compunction meets social dysfunction, health issues and suicidal ideation [1]. It can be associated also with psychosis [2], representing, this latter, the delusional-hallucinatory hypostasis of the emotional atmosphere. The etiopathogenesis of MDD as well as of the other psychiatric disorders, such as Schizophrenia (Scz) and BD are still a challenging forefront of research. The diagnosis of BD-I requires the presence of mania while major depression is not needed. Instead, the *conditio sine qua non* for BD-II is the simultaneous presence of one hypomanic episode (at least) and one major depressive episode [3]. Mania status is marked by hyperactivity, grandiosity ideas, and expansive mood in a psychotic reality interpretation [3]. These pathogenetic pictures are the expression of a neurotransmitter disequilibrium and it is in this framework that the psychiatric relevance of Cbm has to be investigated. In the alveus of the monoamine hypothesis [4,5,6,7], dopamine (DA), norepinephrine (NE), serotonin (5-HT), gamma-aminobutyric acid (GABA) and glutamate (Glu) are the principal actors indicted in the heuristic attempt to trace an etiological identikit of mental disorders [8,9]. Regions enriched with dopaminergic receptors are the striatum (Str); the substantia nigra (SN); the nucleus accumbens (NA), connected to the reward system [10,11,12]; the amygdala (Amg); the hippocampus (Hip); and the frontal cortex [10,13]. Due to the DA involvement in the mechanism of a reward, its reduced activity in the prefrontal cortex (PFC) and the Str is related to anhedonia, a crucial symptom in the clinic manifestation of depression [8,10,12,13]. DA expresses, in fact, its action through two systems: mesolimbic and mesofrontocortical. The mesolimbic system involves NA, Hip and Amg, while the mesofrontocortical pathway organization includes the medial PFC [10,12,13] and anomalies of these tracts have been associated, not only with MDD, but also with Scz and BD [13]. In the brain, norepinephrine is secreted by the locus coeruleus (LC) [5,6] and it has been linked to MDD. Its role seems to be related to the reward system [10] and its production is triggered by stressful stimuli [5]. Serotonin, discharged from the raphe nucleus (RN) [5,10], has surely the leading position in the etiopathogenesis of MDD and BD [14,15,16]. Deficits of 5-HT transporters have been registered in MDD [15,17] and it has been supposed that 5-HT is linked with the severity of the symptoms of these conditions [15]. Among all its receptors (5-HTR), only 5-HT3 receptor (5-HT3R) has captured attention because of its expression in regions linked with the reward system, cognition and depression [14]. Regarding this, there are studies assuming that 5-HT3R antagonists provide anti-depressant effects [10,14]. Although the “monoamine hypothesis” is the most accepted theory, recent works have shown an increasing role played by GABA and Glu in the progression of MDD [4,7,10,18] and BD [19]. GABAergic projections are from the hypothalamus (Hyp) to the occipital and parietal cortex, from Hip to PFC and NA and from the thalamus (Thl) to the Ventral Tegmental Area (VTA) [7,10]. Several works demonstrate a reduction in GABA transporters and in the neurotransmitter itself in subjects diagnosed with MDD and BD [7,10]. Glutamic Acid Decarboxylase (GAD) is an enzyme implicated in the production of GABA; two isoforms, the 65-kDa GAD (GAD65) and the 67-kDa GAD (GAD67) are expressed in human brain [20]. Particularly, cerebellar GAD67 levels are low in patients with MDD and BD [21]. Regarding Glu, its projections start from the PFC to the anterior cingulate cortex, thalamus, VTA, Hip and NA [7,10]. Damage in the production, release and glutamatergic metabolite activity of Glu is shown in MDD patients [7,10]. The down-regulation of Glutamate Transporter 1 (GLT1) can result in a dysregulated metabolism of the neurotransmitter which can be the cause of depressive behaviors [22]. Quite the opposite, if GLT1 is overexpressed, antidepressant outcomes can be observed [22].

The aim of these preliminary notes is to introduce the *status questions*, thus trying to understand if there is the possibility to carve out a diriment role for the Cbm in this theoretical framework.

## 2. Methods

The papers selected for this review focus on the non-motor function of the cerebellum, the neuro-anatomical relations between the different cerebellar areas and the limbic system, as well as on neurochemistry and neuroimaging aspects. Peer-reviewed papers have been included if published in the time frame 1963–2023 and written in English, excluded if inappropriate to the theoretic design of this article and/or focused on conditions not related to mood disorders or emotional perception. Research of studies published from 1963 to 2023 on Scopus, PubMed and Google Scholar databases have been conducted. The results have been transferred to Mendeley and duplicates eliminated. The following is an example of Literature Search performed on 12 February 2023: 1. Cerebellum/2. mood disorders/3. exp MDD/4. exp BD/5. exp emotion/6. neuroimaging/7. Limit to yr = 1963-current 8. Limit to English. All authors screened the chosen articles and discussed the data before writing this review. Data on article characteristics (e.g., year of publication, type of journal, language, titles) have been abstracted and the papers perused on the basis of their pregnancy in mood disorder research, the type of cerebellar area involved, the neuroanatomical connections and impact on emotion processing. After the separation of the duplicates, 945 citations have been identified, whereof 101 were considered eligible. The majority of the studies were published from 2005 to 2023 (*n* = 89) and the most recent papers, from 2018 to 2023, were half of those (*n* = 46). In general, most of the papers share the hypothesis of the Cbm participation in the processing of emotions and in the pathogenesis of MDD and BD.

## 3. Affective Disorders and Cerebellum

The Cbm, anatomically linked to the above-mentioned structures, is notoriously involved in emotional processing [23]. Perception is processed by the thalamic-cortical circuits whose psychological and emotional implications are the result of the activity of the limbic system (LS) [23].

The Cbm is involved in projections with the brainstem and LS, Hyp, septum, Hip, Amg [24] and basal ganglia [9].

Although the precise positioning of the Cbm in the emotion-related systems is still discussed [9,25], the cerebellar pacemaker has been found therapeutic in patients affected by cognitive-behavioral disorders such as Scz, depression and epilepsy [26,27,28]. It has been demonstrated that the correlation between Cbm damage and the impaired capacity of perceiving and then appreciating pleasant emotions [29], on which the Cbm impacts, gives reason for the results obtained by the cerebellar pacemaker in those psychiatric scenarios in which emotional negation dominates.

If, from the one side, the Cbm prominent position is underlined as an *in nuce* coordinator of emotional responses to external stimuli, from the other, it has been noticed the preserved perception of negative emotions even in presence of cerebellar damage. This result can be due, under an evolutionistic profile, to the relevant pregnancy of negative feelings for the purpose of human integrity and survival, whose perceptive capacity cannot be compromised by a unique impairment of Cbm.

The Cbm is characterized by a three-layered cortex and by the cerebellar nuclei, dentate (DN), emboliform, globose and fastigial nucleus (FN), incarnating the crucial crossroads between the cerebellar cortex and the other cerebral structures [30].

They receive inhibitory inputs from the cerebellar cortex and excitatory from the bulbar olivary nuclei [31]. DN projects to the Thl, hence, to the Str, this latter, fundamental in the mesostriatal system, and pivotal in psychiatric disorders.

There are also, highlighted in murine model, dento-amygdala and dento-hypothalamo-amygdala projections [32,33,34]. Medially to DN, emboliform and globose nucleus, interposed nucleus structural articulations, play a key role, together with the FN in movement and balance [35]. In parallel, the Subthalamic Nucleus (StN) is connected by excitatory pathways with the pontine nuclei, projecting, in turn, to the Cbm and, therefore, to the Supplementary Motor Area (SMA). The StN-pontine nuclei-cerebellum-thalamus-cerebral cortex pathway translates into clinical emersion of tremor in Parkinson’s Disease (PD) and a consequent best cognitive safeguard [36,37,38,39]. Functional MRI (fMRI) has, in fact, showed in PD a hyperactivity in StN, and, at the same time, in SMA, as a result of a Cbm-cortical stimulation [38].

These circuits project the Cbm into an articulated network involved in cognitive and emotional functions. This net reflects, biologically, the original logical frame of Cerebellar Cognitive Affective Syndrome (CCAS) [40], a clinical picture resulting from a cerebellar damage characterized by a constellation of neuropsychiatric symptoms [40,41,42,43]: deficit in abstract reasoning and linguistic ability, visuo-spatial disorganization, personality changes, inappropriate behavior and anomia [40,43,44].

In this regard, an interesting compartmentalization has been found: the cognitive symptoms develop in case of lesions mainly in the posterior lobe of the Cbm [44,45], while the affective ones show themselves in associations to damage on the vermis [23,40,46].

Regarding the vermis, neuroimaging technics have documented atrophy in affective and anxiety disorders [30,47]. Speaking of anxiety, not only alterations in volume, but also increased cerebellar blood flow at the level of the superior vermis have been detected [48]. It seems that fear responses, in a cerebellar damage, are preserved, differently from pleasant emotions.

Notably, the Cbm is divided into three zones: **vestibulocerebellum** (vCbm), **pontocerebellum** (pCbm) and **spinocerebellum** (sCbm) [49]. Analyzing the different pathways in the spirit of acquiring a synoptic vision of the cerebellum cognitive-emotional processing is the heart of this work.

## 4. Vestibulocerebellum

The vCbm, also known as Archicerebellum, essentially comprises the flocculonodular lobe. As regards the afferent pathways, it receives fibers from the vestibular nuclei, lower cerebellar peduncles, geniculate nucleus, and superior colliculi. Simultaneously, the axons from this area transmit to the FN and, from there, information goes back to the vestibular nuclei [50] (Figure 1).

Does the vCbm main function relate to motor activity?

According to the hypothesis that would consider the vestibular network as the key to the extra-personal space deciphering that, merging with the internal perception, generates sensory models and anticipation [51], its dysfunction would implicate a mystification of the external environment representation and the sending of what are called “internal fake news” to the Cbm [51].

The processing of this erroneous information, firstly, would generate the implementation of a movement and a behavioral reactivity not appropriate to the surrounding environment, then an altered emotional reaction, since anticipation is fundamental for social interaction [51]. The genesis of a socially “unsuitable” response generates stress that, in terms of chronicity, can participate in the development of a depressive state [51]. A similar interpretation has been provided arguing that the altered perception of the body in relation to the external environment, as in patients with vestibular disorders, is due to the interconnections between the vestibular system, the cerebral cortex and the Hip [52,53]. The disturbed vestibular information translates into “sensory conflicts” that are the basis of the uneven perception of the external world and the resulting distorted and detached behaviors [52]. An inappropriate motor response to the external stimuli, involving the capacity to adapt to the environment, has anxiogenic repercussions on the individual.

Indeed, it has been observed that animal models with otolithic alterations show inability to manage and anticipate their responses when subjected to anxiogenic stimuli [54].

A further hypothesis suggests a connection between posture, equilibrium [55] and anxiety management. In this context, a central role would be played by the parabrachial nucleus, an adjacent area to the superior cerebellar peduncle, involved in the movement and maintenance of position [56]. It has been seen that in rabbits, cerebellar lobule III and flocculus project their cells to this nucleus [56].

The interesting point is that this region has relationships with areas historically involved in emotional processing: the AMG, the infralimbic cortex and the HTH [56,57].

How can the Cbm be involved in managing anxiety? And above all, what role does the Cbm play in the perception of fear? Emotional and behavioral responses to fear are crucial, and their loss has a huge impact on the interactions with other people and reality. In this context, it has been noted that animals with lesions of the vermis are progressively deprived of reactivity to fear stimuli [58]. This represents, however, only a point of intersection, the real basis lies, firstly, in the cerebellar role in the consolidation of memories related to fear and anxiety [59,60]. In this respect, a structured experiment was conducted: first, rats were trained to respond to specific acoustic stimuli, after which, through tetrodotoxin injection, progressive vermis damage was induced [60]. The result led to the gradual loss to the reactivity to the acoustic stimuli for which rats had previously been conditioned [60]. However, the most interesting analysis regarding the experiment was made about the relations between Cbm and Amg. In fact, the greater the acoustic conditioning was in terms of both power and duration, the less the reduction of the response was [60]. The fundamental data is that the total behavioral block occurred if along with the cerebellar damage was inhibited also the Amg, thus suggesting that the two structures are somehow integrated [60]. Once again, it is shown that the CBM cannot be considered an independent structure and that its function also affects ancestral responses, such as reactivity to fear and anxiety [57].

The postulation of a “cognitive-limbic Cbm” about both the vermis (sCbm) [50] and the posterior lobe (pCbm) [50] appears, wherefore, far from the role played by the vCbm. Hence, coming back to the query of this work, is it possible to connect a purely motor cerebellar area, such as the vCbm, to cognitive and emotional experiences?

Reduced functional connectivity between the frontal eye field, Thl, flocculus and ventral paraflocculus, the latter designated for the control of eye movements has been highlighted in subjects diagnosed with BD by resting state functional MRI (rsfMRI) [61].

A further connection between the vCbm, particularly lobule IX and X and BD is the transmembrane protein SLITRK2. This protein, also expressed in the hippocampal dentate gyrus, Str, precerebellar nuclei, and PFC, regulates axonal growth and carries out synaptogenic activity [62]. SLITRK2-mutation-KO mice showed hyperactivity and anti-depressant behaviors, BD clinical features, associated with an increased vestibular activity [62]. In addition, a reduced sensitivity to 5-HT and a failure to respond to therapy with lithium was noted [62].

In light of the relationships that vCbm establishes, it is possible to state that this area plays a fundamental role in the visual-spatial integration and consequently in the bodily self-consciousness [63]. The correct understanding of what surrounds us is substantial in the emotional perception of reality and in the interpretation of social relationships. vCbm represents a research scenario exhibiting, in a magniloquent way, the cerebellar intertwining, to further deepen the motor and behavioral responses to emotional stimuli as the result of complex and indirect brain intersections.

## 5. Pontocerebellum

The pCbm, or Cerebrocerebellum/Neocerebellum, comprises the lateral parts of both cerebellar hemispheres: lobus simplex, superior semilunar Crus I, inferior semilunar lobule Crus II and biventral lobule [41].

Information from contralateral frontal and parietal cerebral lobes arrive to the pontine nuclei [64] and, from there, through the middle cerebellar peduncles, reach the pontocerebellar cortex, where Purkinje cells project to the DN whose fibers extend to the thalamic ventrolateral nucleus through the superior cerebellar peduncles, thus reaching the primary motor cortex (PMC). This is the cortico-pontine-thalamic-cortical loop [64,65].

The pCbm main function is insofar related to planning movements that are about to happen, perfecting their precision [41]. Nevertheless, since frontal and posterior parietal cortices are crucial in cognitive functions, it is reasonable to postulate the possibility that Cbm also contributes to the emotional and cognitive processing [41,64].

The model of a pCbm, endowed with polyhedric functions is supported by the variety of connections with cortical structures: the PMC, for instance, sends information to the lobules IV, V and VI; while inputs from the Brodmann area 46 in the dorsolateral PFC reach Crus I and Crus II [64,66,67,68].

Furthermore, a study using rsfMRI conducted in monkeys and cats has proved the existence of a “bilateral limbic-cerebellar-amygdaloid network” consisting of connections among the dentate nuclei, the hemispheric part of lobule VIII–IX and Amg [69,70] (Figure 2).

By using fMRI, it has been noticed the activation of Crus I and Crus II, not only during cognitive processes, such as language, working memory and decision-making, but also subsequent to emotional stimuli [45]. Regarding the experience of pain, a distinction has to be made. When facing our own pain, the stimulated cerebellar region is the vermis, nevertheless when it comes to feeling empathy for someone else’s pain, it is the lobule VI of pCbm that turns on [45]. Moreover, research focused on the intrinsic connectivity networks, using MRI, has demonstrated that both DN and pCbm undergo a powerful stimulation during cognitive and emotional processes [70,71].

These proofs have given significant steps forward in extricating the Cbm from the mere motor function. It is possible to assume that the “cognitive Cbm” coincides not only with the pCbm highlighting, consequently, that all the three cerebellar zones are, at different levels, involved in higher-cognitive pathway [44,72,73].

Numerous studies have showed how the Cbm is activated during social, mnestic activities and emotional events [41,74].

With the aim of connecting Cbm with MDD, a study was carried out by analyzing differences between two groups, one, with patients diagnosed with geriatric depression, the other as health control [42]. The results brought to light the correlation between vermis-posterior cingulate cortex pathway and the severity of MDD symptoms [42]. The first group exhibited, in fact, an increase in vermal blood-flow and a reduction of the general volume in MDD [42,75].

Moreover, it was demonstrated that medial PFC and anterior cingulate are functionally linked with Crus I [42]. RsfMRI studies in subjects with MDD have showed reduced functional connectivity between Crus I and the supramarginal gyrus and between Crus II and the angular gyrus [76]. These two cerebral convolutions belong to the parietal lobe and, in addition to being involved in language perception, have a role in controlling and managing emotional attentional processes [76]. The angular gyrus also establishes connections with the flocculus [63].

Staying in the geriatric field, 3-T MRI was performed on a cohort resulting in what is called “physio-cognitive decline syndrome” (PCDS) in 15.9% of the participants [77]. The mechanism at the base of PCDS was confirmed to be a neurocircuit between left Cbm lobules VI and V and Hip-Amg [46,77,78]. Furthermore, it was highlighted that this syndrome is correlated with a decreased volume in limbic structures, the cerebral cortex and the Cbm itself [77].

Considering that it is well known that Amg is implicated in emotive integration and that in Hip lies the “emotional memory” [46], these evaluations strongly put in evidence the neuroanatomic involvement of pCbm in emotional processes and dysregulation. With the purpose of endeavoring monosynaptic connections between the Cbm and the Amg, an anterograde tracer virus was injected into the deep cerebellar nuclei, while a retrograde one was injected into the basolateral Amg [74]. Monosynaptic circuits between the two structures were not found. However, a new decisive disynaptic circuit between the deep cerebellar nuclei and the basolateral Amg, in which the fundamental junction between the two components is represented by the Thl, has been identified [74]. This finding represents a crucial step in understanding how the Cbm fulfils functions that go beyond the purely somato-motor ones. It can be assumed that, somehow, scientists are looking for the anatomical basis of what is basically called “non-verbal communication”.

Furthermore, it has been noticed that the stimulation of posterior cerebellar lobe through electrodes can mitigate depressive symptoms, especially in MDD subjects whose astrocytes produce less Glial Fibrillary Acidic Protein (GFAP), not observed in people with BD [43].

Regarding BD, besides an abnormal functionality of the anterior LS, molecular mechanisms in the Cbm have been proposed [43]. Immunohistochemistry and molecular analysis on proteins implicated in plasticity such as BDNF and its receptor tyrosine kinase B (TrkB) in patients with MDD, BD have been investigated [79]. Modification of these molecules was substantial especially at the level of Crus II [79].

In particular, the BD group showed a reduction in the expression of TrkB [79]. It has been suggested that the isoform TrkB-T1 can diminish Brain-Derived Neurotrophic Factor (BDNF) capacity to stimulate neural growth and survival and therefore modify synapsis plasticity and lead to alteration of astrocytes [79].

Starting from the pCbm and widening now the discussion to the whole Cbm, further molecular alterations have been described. The first concerns Transcription Factor SP4, whose gene has been associated with BD, the second refers to modifications of mitochondrial genes in BD and MDD [79]. SP4 is strongly expressed in Cbm and Hip neurons and promotes neuronal development [79]. The reduced presence of the protein in the post-mortem cerebellar tissue of subjects with BD has given rise to hypotheses linking this mood disorder to SP4 [79]. It has been hypothesized that lithium can control SP4 itself, reducing its expressiveness [79]. This point is interesting as it represents a possible change in the treatment of BD. In subjects diagnosed with BD, on the one hand there is the reduction of mitochondrial genes such as NDUFV1 and NDUFV2, at the level of the Hip and PFC. On the other hand, there is an increase in these genes’ expression in the parietal and occipital cortex [79]. At the cerebellar level, however, their expression is decreased [79]. Focusing on other proteins, astrocytes located near vessels express aquaporin-4 (AQP4) which is a water-selective membrane transport protein [80]. Its apparent diffusion coefficient (ADCuh) is augmented in cerebellar hemispheres and DN and the level of the depressive status impacts on ADCuh [80]. In post-mortem samples AQP4 is upregulated in the prefrontal lobe of BD subjects, while it is downregulated in the locus coeruleus of MDD [80]. The pCbm represents, therefore, the “cognitive” component of the Cbm: its activation during cognitive processes and its associations with areas involved in cognitive, emotional and mnestic function show that the cerebellar hemispheres are active protagonists in the behavioral reaction to emotional stimuli.

## 6. Spinocerebellum

The sCbm, or Paleocerebellum, corresponds to the vermis and the intermediate parts of the hemispheres known as “paravermis”. It receives somatosensory inputs from the spinocerebellar tract and the pontine reticular-tegmental nucleus. Paravermis Purkinje cells make synapses on the interposed nuclei; from there information goes to red nucleus where, after making a decussation, the lateral descending systems start [65]. Vermis Purkinje cells, instead, project their GABAergic axons [71] to the FN that sends inputs to the reticular formation, from where, through a decussation, the medial descending tracts begin, and to the PMC, passing through the ventral lateral nucleus of the Thl [65]. In addition to these pathways, vermis is a key juncture in processing emotional inputs: it is for this reason that the so-called “limbic Cbm” is identified with this area [45,70,81]. In relation to this, it has been observed, in subjects with BD, a functional connectivity between the vermis, in particular the lobules V, VIIIb and X, and anatomical regions notoriously belonging to the LS, such as the postcentral gyrus, the cingulate gyrus and the Amg [46,82]. Furthermore, lobule VIIIb, along with VIIb, organizes direct connections with the Thl [82]. Even the Hyp is indirectly linked to sCbm: Hyp projections, in fact, through the medial portions of the pons, reach the vermis [23] (Figure 3).

This system has been interpreted as a mechanism of integration of somatic and limbic inputs made by the Cbm [23]. Already in 1970, the vermis was conceived as somehow involved in emotional-behavioral functions: in this regard, the stimulation by electrodes of its cortex was exploited to reduce the symptomatology of epileptic subjects [83]. Unexpectedly, this attempt also resulted in an improvement of depressive symptoms, anger, and aggression [83].

Based on these anatomical assumptions, the role played by the pCbm in MDD and BD was subsequently investigated. Furthermore, in remission-MDD patients, treated with electroconvulsive therapy, showed a subsequent increased volume of lobule VIIA [84], which has a particular relevance because it is not connected to any somato-motor component [85]. Some studies, however, have focused not only on cerebellar volume, but also on the perfusion of specific brain regions.

MDD and BD subjects show reduced blood flow in the left anterior cingulate and the left dorsolateral PFC, offset by an increased flow to the vermis [86,87,88]. It has been assumed that depressive or manic states develop precisely when this cerebellar “compensation” fails [82].

In this regard, posterior-vermis lesions or dysfunctions may be the key to the blooming of psychotic characteristics, such as hallucinations [82]. Compensation explicated not only in an increased perfusion, but also in an augmented metabolism of the Cbm and Thl in response to hypo-metabolism of the insula, LS and basal ganglia [89]. The increased volume of the posterior vermis, especially regions VI and VIII, has been associated with the severity of depressive symptoms [90,91], while it has been shown that an increased volume of the anterior vermis occurs in MDD subjects treated for years [92]. In BD subjects, vermis shows reduced resting-state functional connectivity (rsFC) with the ventral PFC; a decrease also seen in connectivity between the anterior vermis and the cingulate cortex [81,93].

Patients with MDD show reduced functional connectivity between the Lobule IX and the Insula, a brain area well known for its affecting emotions and empathy [76]. Normally the insular cortex participates in the positive perception of emotions. When the connections between the insula, cerebral cortex and Cbm are altered, emotions have a lesser positive impact on mood that tends, therefore, to subside [76]. Along with the insula, an additional brain area fundamental to the affective sphere is represented by the anterior cingulate cortex [94]. In this regard, it has been observed that patients with MDD exhibit a volumetric reduction of the grey matter at the level of this area, unlike those diagnosed with BD in which this decrease is evidenced in the Hip and Amg [94]. At the cerebellar level, the anterior cingulate cortex establishes relationships with both Crus I and Crus II, and in case of BD, these connections show reduced functional connectivity by confirming, once again, that the role of the Cbm cannot be limited to motor functions [94].

A relationship that surely must be examined in depth is between the pCbm and Amg. The connectivity of the latter can be even used to distinguish functionally depressive states from manic ones [90,95]. At the same time its volume can differentiate an untreated MDD, where the volume has increased, from the treated one, where the volume is reduced [95]. Interestingly, in BD the situation is diametrically opposite: in untreated subjects the volume of the Amg is reduced, while it is increased in those who follow a therapy [95].

Vermis, in relation to BD, has also been studied under a neurochemical profile.

The levels of N-acetylaspartate (NAA), myo-inositol and choline were analyzed in the cerebellar vermis in children familiar with BD who began to show mood disorders without resulting in manic episodes [96]. Participants showed a reduction of all three neurometabolites [96,97]. It was also noted that lithium was able to increase brain levels of Myo-inositol [96] and NAA [98], thus confirming its therapeutic value.

If the vCbm and pCbm represent, respectively, the “perceptive” and the “cognitive” component, to the sCbm is assigned perhaps the most representative function of the emotional processes: the “limbic” one. By virtue of its relationships with the limbic system and the volumetric and functional changes it encounters in subjects diagnosed with mood disorders, sCbm conquers a privileged place in the definitive understanding of the anatomo-functional mechanisms of emotions.

## 7. Conclusions

For decades, the Cbm has been considered the dominant zone for the integration, improvement and implementation of motor activity. However, thanks to modern neuroimaging and neurochemical techniques it has been possible to expand its role to more complex domains.

In this regard, the social impact of mood disorders and the arduous understanding of their pathogenesis have led to the search for the contribution made by the intricate cerebellar mechanisms in the development of such conditions. The understanding of cerebellar topography related to the processing of emotions has grown over time, however gaps on what is the exact role of the Cbm in mood disorders remain. Studies have demonstrated that there are anomalies in cerebellar metabolism and cellular activity. Even so, literature lacks active research on the impact of those anomalies on neuropsychiatric symptoms.

The initial reference to mental illness, in particular to depression as a disorder of the social dimension of the individual, has given way to an analysis on the role of the Cbm not only in affective disorders, but also in building social relationships.

The Cbm, in fact, develops internal models making coordination and movement execution more precise, working in concert with the cerebral cortex. The Cbm anticipates action consequences [99], assesses movement appropriateness with respect to the context [100], and, consequently, affects behavior as the combination of movement and cognition in a given situation [40].

To sum up, each cerebellar zone participates, at different levels, in perception and emotional processing, performing tasks of anticipation, integration and processing of emotional perceptions crucial in mood stability. When the balance of the numerous cerebro-cerebellar connections is disturbed, a discrepancy between the reality surrounding us and its perception is created. The misinterpretation of emotions results in a “mismatch” amplifying the effect of negative emotions on mood and lack of control in executing behaviors aimed at correcting the emotional representation of reality [101].

The analysis of cerebellar activity proves to be increasingly stimulating, as the Cbm can epitomize not only a fundamental turning point for the understanding of affective disorders, then a potential innovative target for their treatment, but also a crucial protagonist in the individual harmonic social integration.

## Figures and Tables

**Figure 1 metabolites-13-01113-f001:**
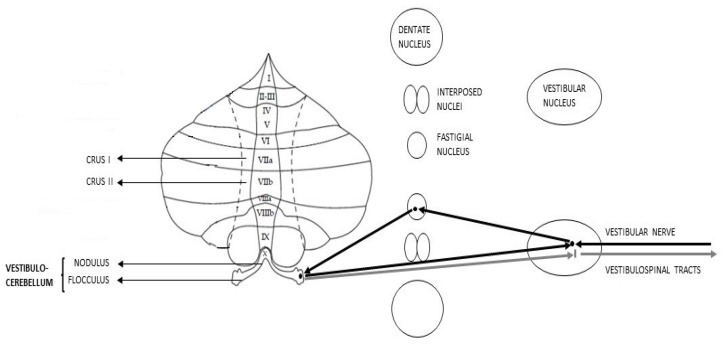
Schematic representation of Vestibulocerebellar connections (afferent pathways in black, efferent pathways in grey).

**Figure 2 metabolites-13-01113-f002:**
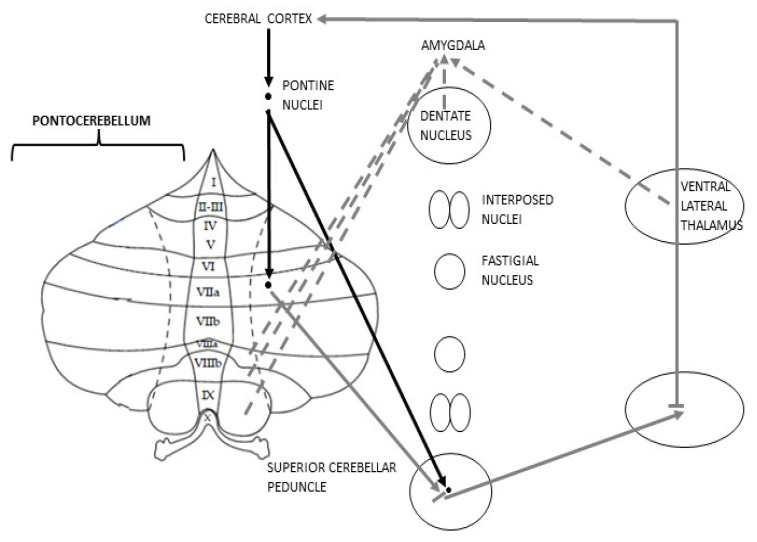
Schematic representation of Pontocerebellar connections; discontinuous efferences represent the anatomical rationale for the postulation of the cognitive Cbm.

**Figure 3 metabolites-13-01113-f003:**
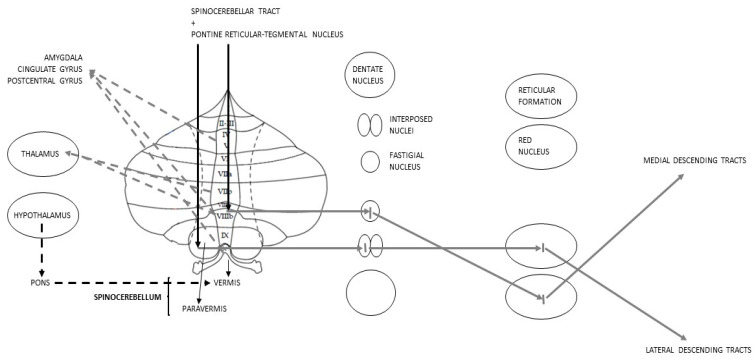
Schematic representation of Spinocerebellar connections; discontinuous lines represent the anatomical rationale for the postulation of the limbic Cbm.

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
