# Peer review of "The Cerebellum’s Role in Affective Disorders: The Onset of Its Social Dimension"

_metabolites, 2023, doi:10.3390/metabo13111113_

Round 1

Reviewer 1 Report

Comments and Suggestions for Authors

This is a narrative review on the role of cerebellum in affective disorders. It delivers interesting information based on 97 literature sources. The paper may be considered for publication after major revision. One critical point would be to specify the search strategy, inluding key words, scientific data bases, time frame. The emphasis should rest on publications from the last 5 years. The other point would be to more carefully address recent studies on functional connectivity of cerebellum with other brain regions, e.g. anterior cingulate cortex (ACC). On that instance authors may find it interesting (but not mandatory at all) to refer to: https://doi.org/10.3390/biomedicines11061608

Reviewer 2 Report

Comments and Suggestions for Authors

This comprehensive review extensively explores the potential roles of the cerebellum, traditionally associated with motor control, in cognitive and emotional processing. The article takes a comprehensive view by examining the three functional zones of the cerebellum and their connections to brain regions involved in emotion. It also establishes their relationships with depression and bipolar disorder. Supported by a wealth of neuroimaging and neurochemical research, the review reveals abnormalities in cerebellar volume, blood flow, and functional connectivity in patients with depression and bipolar disorder. The author effectively correlates this statistical evidence with clinical symptoms, thereby highlighting the potential roles of cerebellar abnormalities in the pathogenesis of these disorders.

My major concern is with the abstract, which is too vague and does not clearly state the central point of this manuscript - the potential role of the cerebellum in the pathogenesis of depression and bipolar disorder. There is no mention of the three functional zones of the cerebellum and their connections to brain regions involved in emotion processing, which are crucial aspects of this manuscript. It does not adequately summarize the research findings on cerebellar volume and functional abnormalities in patients with depression and bipolar disorder. Overall, the abstract should be more concise and focused, highlighting the author's innovative perspectives and research contributions.

I recommend revising the abstract to clearly state the research objectives and emphasize the key findings - evidence of structural and functional abnormalities in the three zones of the cerebellum in depression and bipolar disorder, supporting the cerebellum as a potential new therapeutic target for these disorders.

Another concern is that the article extensively describes the structure of the cerebellum and its connections with other brain regions in various sections, but it does not effectively highlight the relevance of these connections to mental disorders. When discussing the three functional areas of the cerebellum, it should emphasize their connections with emotional brain regions in plain language, rather than just describing anatomical structures. Please provide more detailed information on the connections between the cerebellum and mental disorders.

Overall, the article's well-structured organization and natural paragraph transitions result in fluent language expression. The extensive and diverse list of references enhances the persuasiveness of the arguments. And this review provides a profound, comprehensive, and engaging exploration of the cerebellum's cognitive-emotional functions. It offers valuable insights into the cerebellum's potential roles in the pathogenesis of mental disorders, making it a significant contribution to this field.

Round 2

Reviewer 1 Report

Comments and Suggestions for Authors

The authors have considered carefully all comments from peer review report. The paper may be accepted in its current, revised form.